Use and effectiveness of dapagliflozin in patients with type 2 diabetes mellitus: a multicenter retrospective study in Taiwan

Chen Jung-Fu 1 2
Peng Yun-Shing 3
Chen Chung-Sen 4
Tseng Chin-Hsiao 5
Chen Pei-Chi 6
Lee Ting-I 7 8
Lu Yung-Chuan 9 10
Yang Yi-Sun 11
Lin Ching-Ling 8 12
Hung Yi-Jen 13
Chen Szu-Ta 14
Lu Chieh-Hsiang 15 16
Yang Chwen-Yi 17
Chen Ching-Chu 18 19
Lee Chun-Chuan 20
Hsiao Pi-Jung 9
Jiang Ju-Ying 21
Tu Shih-Te 10836@cch.org.tw 22
1 Division of Endocrinology and Metabolism, Department of Internal Medicine, Kaohsiung Chang Gung Memorial Hospital , Kaohsiung , Taiwan
2 College of Medicine, Chang Gung University , Taoyuan , Taiwan
3 Division of Endocrinology and Metabolism, Department of Internal Medicine, Chiayi Chang Gung Memorial Hospital , Chiayi , Taiwan
4 Division of Endocrinology and Metabolism, Department of Internal Medicine, Taipei City Hospital Zhongxiao Branch , Taipei , Taiwan
5 Division of Endocrinology and Metabolism, Department of Internal Medicine, National Taiwan University Hospital , Taipei , Taiwan
6 Division of Endocrinology and Metabolism, Department of Internal Medicine, Shin Kong Wu Ho-Su Memorial Hospital , Taipei , Taiwan
7 Division of Endocrinology and Metabolism, Department of Internal Medicine, Wan Fang Hospital, Taipei Medical University , Taipei , Taiwan
8 Department of General Medicine, School of Medicine, College of Medicine, Taipei Medical University , Taipei , Taiwan
9 Division of Endocrinology and Metabolism, Department of Internal Medicine, E-Da Hospital , Kaohsiung , Taiwan
10 School of Medicine, College of Medicine, I-Shou University , Kaohsiung , Taiwan
11 Division of Endocrinology and Metabolism, Department of Internal Medicine, Chung Shan Medical University Hospital , Taichung , Taiwan
12 Division of Endocrinology and Metabolism, Department of Internal Medicine, Cathay General Hospital , Taipei , Taiwan
13 Division of Endocrinology and Metabolism, Department of Internal Medicine, Tri-Service General Hospital Songshan Branch , Taipei , Taiwan
14 Division of Endocrinology and Metabolism, Department of Internal Medicine, Linkou Chang Gung Memorial Hospital , Taoyuan , Taiwan
15 Division of Endocrinology and Metabolism, Department of Internal Medicine, Chia-Yi Christian Hospital , Chia-Yi , Taiwan
16 Lutheran Medical Foundation, Kaohsiung Christian Hospital , Kaohsiung , Taiwan
17 Division of Endocrinology and Metabolism, Department of Internal Medicine, Chi Mei Medical Center , Tainan , Taiwan
18 Division of Endocrinology and Metabolism, Department of Medicine, China Medical University Hospital , Taichung , Taiwan
19 School of Chinese Medicine, China Medical University , Taichung , Taiwan
20 Division of Endocrinology and Metabolism, Department of Internal Medicine, Mackay Memorial Hospital , Taipei , Taiwan
21 Division of Endocrinology and Metabolism, Department of Internal Medicine, Far Eastern Memorial Hospital , New Taipei City , Taiwan
22 Division of Endocrinology and Metabolism, Department of Internal Medicine, Changhua Christian Hospital , Changhua , Taiwan
Foti Daniela
Electronic publication date: 2020 Nov 17
Publication date: 2020
Volume: 8
Electronic Location ID: e9998
Received 2020 Mar 6; Accepted 2020 Aug 28
Copyright: ©2020 Chen et al.
Copyright year: 2020
Copyright holder: Chen et al.
License: This is an open access article distributed under the terms of the Creative Commons Attribution License, which permits unrestricted use, distribution, reproduction and adaptation in any medium and for any purpose provided that it is properly attributed. For attribution, the original author(s), title, publication source (PeerJ) and either DOI or URL of the article must be cited.
License URL: https://creativecommons.org/licenses/by/4.0/

Keywords: Dapagliflozin, HbA1c, SGLT2 inhibitors, Type 2 diabetes mellitus, Real-world evidence

Funding: AstraZeneca Taiwan, Ltd The study was sponsored by AstraZeneca. Writing and editorial assistance was also funded by AstraZeneca Taiwan, Ltd. The funders had no role in study design, data collection and analysis, decision to publish, or preparation of the manuscript.

==============================
Aims/Introduction

To investigate the clinical outcomes of patients with type 2 diabetes mellitus (T2DM) who initiated dapagliflozin in real-world practice in Taiwan.

Materials and Methods

In this multicenter retrospective study, adult patients with T2DM who initiated dapagliflozin after May 1st 2016 either as add-on or switch therapy were included. Changes in clinical and laboratory parameters were evaluated at 3 and 6 months. Baseline factors associated with dapagliflozin response in glycated hemoglobin (HbA1c) were analyzed by univariate and multivariate logistic regression.

Results

A total of 1,960 patients were eligible. At 6 months, significant changes were observed: HbA1c by −0.73% (95% confidence interval [CI] −0.80, −0.67), body weight was -1.61 kg (95% CI −1.79, −1.42), and systolic/diastolic blood pressure by −3.6/−1.4 mmHg. Add-on dapagliflozin showed significantly greater HbA1c reduction (−0.82%) than switched therapy (−0.66%) (p = 0.002). The proportion of patients achieving HbA1c <7% target increased from 6% at baseline to 19% at Month 6. Almost 80% of patients experienced at least 1% reduction in HbA1c, and 65% of patients showed both weight loss and reduction in HbA1c. Around 37% of patients had at least 3% weight loss. Multivariate logistic regression analysis indicated patients with higher baseline HbA1c and those who initiated dapagliflozin as add-on therapy were associated with a greater reduction in HbA1c.

Conclusions

In this real-world study with the highest patient number of Chinese population to date, the use of dapagliflozin was associated with significant improvement in glycemic control, body weight, and blood pressure in patients with T2DM. Initiating dapagliflozin as add-on therapy showed better glycemic control than as switch therapy.

Introduction

Type 2 diabetes mellitus (T2DM) is a chronic metabolic disease affecting populations worldwide, and it has become an important public health challenge among Asian countries, especially in ethnic Chinese populations (Cho et al., 2018; Lim & Chan, 2019; Lim et al., 2017; Nanditha et al., 2016). According to the 2019 IDF Diabetes Atlas and a nationwide database analysis in Taiwan, the prevalence of diabetes increase from 4.31% in 2000 to 6.6% in 2019 for adults aged 20–79 years, resulting in a more than 70% increase in the total diabetic population (1.23 million in 2019) (International Diabetes Federation, 2019; Jiang et al., 2012). Currently, most guidelines recommend pharmacologic therapy based on evaluating glycated hemoglobin (HbA1c) levels for glycemic control. When the glycemic target is not achieved by lifestyle management and metformin, a second agent may be initiated, considering medication profiles and patient-related factors (American Diabetes Association, 2019; Diabetes Association Of The Republic Of China Taiwan, 2019; Garber et al., 2018; McGuire et al., 2016).

Sodium-glucose cotransporter-2 (SGLT2) inhibitors are a newer class of oral antidiabetic drugs (OADs) that inhibit glucose reabsorption at the early segments of the proximal convoluted tubule, thereby promoting glucosuria independently of insulin action (Hasan, Alsahli & Gerich, 2014). These agents improve glycemic control without increasing the risk of hypoglycemia, and have pleiotropic effects such as weight loss and reduction in blood pressure (BP). Combining SGLT2 inhibitors with metformin has been demonstrated to have additive effect compared to metformin alone in HbA1c and body weight reduction (Molugulu et al., 2017). Given that T2DM has been known to have a higher risk of cardiovascular events, significant attention has been paid to the benefit of SGLT2 inhibitors on cardiovascular outcomes in T2DM patients with or without pre-existing cardiovascular disease (Scholtes et al., 2019).

Dapagliflozin, a member of the SGLT2 inhibitor class, has been shown in randomized controlled trials (RCTs) to improve glycemic control both as monotherapy (Bailey et al., 2015) and as add-on to other antidiabetic drugs, (Nauck et al., 2014; Sun et al., 2014) along with well-established safety profile in a pooled analysis of phase IIb/III trials (Jabbour et al., 2018). Complimentary to RCTs, observational studies can provide real-world data reflecting clinical practice patterns and outcomes not collected in RCTs (Garrison Jr et al., 2007).

The real-world evidence on dapagliflozin has been reported in North America (Brown, Gupta & Aronson, 2017; Huang et al., 2018b) and Europe, (Fadini et al., 2018; Mirabelli et al., 2019; Scheerer et al., 2016; Scorsone et al., 2018; Wilding et al., 2017) whereas data are limited in Asia, especially for ethnic Chinese populations (Han et al., 2018; Tobita et al., 2017; Viswanathan & Singh, 2019). Dapagliflozin has been reimbursed by the National Health Insurance (NHI) in Taiwan since May of 2016. The purpose of the LEAD (Learning from RWE: A Multicenter retrospective study of Dapagliflozin in Taiwan) study was to retrospectively investigate the clinical outcomes for patients with T2DM initiating dapagliflozin under real-world setting in Taiwan.

Materials & Methods

Data source and study population

This was a multicenter retrospective observational study (ClinicalTrials.gov identifier NCT03084965) enrolling patients with T2DM exposed to dapagliflozin after dapagliflozin reimbursement in Taiwan since May 2016. Medical information was extracted manually by chart review and recorded in a standardized case report form after obtaining written informed consent from the patients. The final protocol of this study, including the final version of the subject informed consent form, had been approved by the Ethics Committee/Institutional Review Board (IRB)/Independent Ethics Committee (IEC) of each sites (Institutional Review Board Changhua Christian Hospital, Research Ethics Committee China Medical University & Hospital, The Institutional Review Board Chung Shan Medical University Hospital, Institutional Review Board Chia-Yi Hospital, Taipei City Hospital Research Ethics Committee, Chang Gung Medical Foundation Institutional Review Board, Kaohsiung Medical University Chung-Ho Memorial Hospital, Institutional Review Board Chi Mei Medical Center, Institutional Review Board of the E-DA Hospital, The Institutional Review Board of Shin Kong Wu Ho-Su Memorial Hospital, Institutional Review Board, Tri-Service General Hospital, Research Ethics Review Committee Far Eastern Memorial Hospital, Institutional Review Board of the Cathay General Hospital, TMU-Joint Institutional Review Board, National Taiwan University Hospital Research Ethics Committee, Institutional Review Board MacKay Memorial Hospital).

Patients were eligible for inclusion if they (1) were diagnosed with T2DM and aged ≥20 years old, (2) initiated 5 mg or 10 mg of dapagliflozin after May 1st 2016, either as add-on therapy to existing OAD(s) with or without insulin, or as switch therapy from another OAD, and (3) completed follow-up of at least 6 months regardless of continuation on dapagliflozin therapy. Data were excluded if they (1) received other SGLT2 inhibitors prior to the initiation of dapagliflozin, (2) had a diagnosis of type 1 diabetes, or (3) were included in other clinical trials concurrently during the retrospective data collection period.

Assessment and outcome measures

Patient baseline demographics and clinical characteristics were recorded at the time of dapagliflozin initiation. Reasons for starting/switching to dapagliflozin, rate and reasons of dapagliflozin discontinuation were collected. Measurements of HbA1c, body weight, BP, fasting plasma glucose (FPG), and lipid profile (total cholesterol, high-density lipoprotein cholesterol [HDL-C], low-density lipoprotein cholesterol [LDL-C], and triglycerides) were obtained at baseline, 3 and 6 months to assess for changes. Further subgroup analyses were performed to assess changes in HbA1c by baseline HbA1c, manner of dapagliflozin initiation (as add-on or switch), BMI, age, and dosage of dapagliflozin. Changes in body weight were also analyzed by subgroups of baseline BMI, HbA1c, age, and dosage of dapagliflozin. Patients who have one or more times record using 5 mg dapagliflozin during the follow up were defined as 5 mg group. The proportion of patients within each glycemic level (HbA1c <7%, 7–8%, 8–9%, and >9%) was evaluated at baseline and 6 months.

Statistical analysis

Descriptive statistics were provided for all variables. Continuous variables were presented with mean ± standard deviation, and numbers or percentages were used for categorical variables. Changes from baseline of the variables were analyzed using evaluable data at the respective time points. Mean changes of each variable from baseline to Month 3, baseline to Month 6, and between 3 and 6 months were evaluated by paired t-test and Wilcoxon signed-rank test in the overall cohort and subgroups. Differences in HbA1c reduction between dapagliflozin add-on and switch groups at Month 3 and Month 6 were analyzed by Wilcoxon rank-sum test. For changes in HbA1c and body weight among other subgroups at Month 3 and Month 6, the Kruskal-Wallis test was used. Patient distributions of changes in HbA1c and body weight at Month 6 were displayed by scatter plots for total, add-on, and switch groups. Baseline factors associated with dapagliflozin response in HbA1c were examined by univariate and multivariate logistic regression analyses. The cutoff for dapagliflozin responders was determined by using the median of change in HbA1c in patients with evaluable data at Month 6. Regarding the missing data, patients without the respective data were excluded in each analysis. For instance, when analyzing change from baseline to Month 3, only patients with both baseline and Month 3 data were included. Statistical significance was set at p value <0.05. All statistical analyses were conducted using SAS version 9.3.

Results

Baseline characteristics

A total of 1960 patients were eligible, with a mean age of 57.8 ± 11.5 years, and 52% were male. The baseline clinical and laboratory characteristics are shown in Table 1. The mean HbA1c was 8.8 ± 1.4%, and body weight was 75.2 ± 15.8 kg. Baseline systolic blood pressure (SBP) and diastolic blood pressure (DBP) was 136.6 ± 17.6 and 77.8 ± 11.3 mmHg, respectively. Metformin was the most prescribed antihyperglycemic agent (92.4%), followed by sulfonylurea (70.7%) and dipeptidyl peptidase-4 (DPP4) inhibitor (53.5%). Over half of the patients were on either dual (38.2%) or triple (36.2%) therapy prior to initiating dapagliflozin. A large proportion of patients (78.9%) initiated dapagliflozin at the higher dose (10 mg). Regarding the manner of initiation, more than half (53.8%) of the patients were switched to dapagliflozin, while the others (46.2%) started dapagliflozin as add-on therapy. Main reasons for starting dapagliflozin were for its HbA1c lowering efficacy (81.2%), followed by less concern for weight gain (26.8%) and lower risk of hypoglycemia (11.4%).

Table 1 Baseline characteristics in 1,960 patients.

Age (years)	57.8 ± 11.5	
Sex (male)	1020 (52%)	
Weight (kg)	75.2 ± 15.8	
BMI (kg/m2)	28.3 ± 4.9	
BMI categories		
<24	17.6%	
≥24 –<27	26.0%	
≥27 –<30	24.9%	
≥30 –<35	22.3%	
≥35	9.2%	
HbA1c (%)	8.8 ± 1.4	
HbA1c distribution	
<7%	5.8%	
≥7% –<8%	25.1%	
≥8% –<9%	31.4%	
>9% –≤10%	19.6%	
>10%	18.1%	
FPG (mg/dL)	173.0 ± 53.9	
SBP (mmHg)	136.6 ± 17.6	
DBP (mmHg)	77.8 ± 11.3	
Total cholesterol (mg/dL)	164.3 ± 36.0	
Triglycerides (mg/dL)	171.2 ± 133.6	
LDL-C (mg/dL)	89.8 ± 27.8	
HDL-C (mg/dL)	44.8 ± 13.5	
Medical history	
Hypertension	60.5%	
CAD	7.8%	
Cerebrovascular disease	1.7%	
PAD	7.4%	
Heart failure	2.6%	
Nephropathy	34.6%	
Retinopathy	10.7%	
Neuropathy	14.1%	
Antihyperglycemic therapy	
Metformin	92.4%	
Sulfonylurea	70.7%	
DPP4 inhibitor	53.5%	
Meglitinide	3.0%	
α-glucosidase inhibitor	16.7%	
Thiazolidinedione	26.6%	
GLP-1 RA	3.0%	
Insulin	29.5%	
Number of antihyperglycemic drug prior to dapagliflozin	
Monotherapy	11.9%	
Dual therapy	38.2%	
Triple therapy	36.2%	
>3 drugs combination	13.8%	
Manner of dapagliflozin initiation	
Add-on	905 (46.2%)	
Switch	1055 (53.8%)	
Dosage of dapagliflozin	
Dapagliflozin 5 mg	413 (21.1%)	
Dapagliflozin 10 mg	1547 (78.9%)	
Antihypertensive therapy	61.7%	
ACEI or ARB	49.6%	
β-blocker	21.1%	
Calcium channel blocker	30.9%	
Diuretic	11.3%	
Others	3.0%	
Lipid-lowering therapy	74.5%	
Statin	68.6%	
Fibrate	9.7%	
Niacin	2.3%	
Ezetimibe	9.6%	
Antithrombotic therapy	
Antiplatelet	22.5%	
Anticoagulant	2.1%	
Notes.

Values are in n (%), mean ± SD, or percent of total population. The following parameters had patients (n) with missing data: Weight, n = 192; BMI, n = 206; HbA1c, n = 119; FPG, n = 380; SBP/DBP, n = 210; Total cholesterol, n = 910; Triglycerides, n = 772; HDL-C, n = 1052; LDL-C, n = 538.

ACEI angiotensin-converting-enzyme inhibitors

ARB angiotensin II receptor blockers

BMI body mass index

CAD coronary artery disease

DBP diastolic blood pressure

DPP4 dipeptidyl peptidase-4

FPG fasting plasma glucose

GLP-1 RA glucagon-like peptide-1 receptor agonists

HbA1c glycated hemoglobin

HDL-C high-density lipoprotein cholesterol

LDL-C low-density lipoprotein cholesterol

PAD peripheral artery disease

SBP systolic blood pressure

Effectiveness of dapagliflozin on clinical and laboratory parameters

Table 2 shows changes in clinical and laboratory parameters after dapagliflozin initiation at 3 and 6 months. Compared with baseline, statistically significant reductions in HbA1c were observed both at Month 3 (−0.68%; 95% confidence interval [CI] -0.74, −0.62, p < 0.001) and at Month 6 (−0.73%; 95% CI −0.80, −0.67, p < 0.001). Considering patients with evaluable HbA1c data at baseline and Month 6, the proportion of patients achieving the glycemic target (HbA1c <7%) was 6% at baseline, and it increased to 19% by Month 6 after initiating dapagliflozin. Moreover, the proportion of patients who were poorly controlled (HbA1c >9%) decreased from 34.7% at baseline to 15.9% at Month 6 (Fig. S1A). Improvements in FPG (−28.3 mg/dL), BMI (-0.60), body weight (−1.61 kg), and SBP/DBP (−3.6/−1.4 mmHg) at Month 6 were also significant (all p < 0.001, Table 2). Aside from BP reduction that plateaued at Month 3, small but significant improvements were observed in HbA1c, FPG, BMI, and body weight from Month 3 to Month 6. Small changes in lipid profiles were also noted (Table 2).

Table 2 Changes in clinical and laboratory outcomes from baseline to 3 and 6 months.

Parameter	Month 3	Month 6	Month 3 versus Month 6	
	n	Change	pvalue	n	Change	pvalue	n	Change	pvalue	
HbA1c, %	1344	−0.68 (−0.74, −0.62)	<0.001	1197	−0.73 (−0.80, −0.67)	<0.001	1094	−0.07 (−0.11, 0.02)	0.001	
BMI, kg/m2	1147	−0.50 (−0.56, −0.45)	<0.001	1006	−0.60 (−0.67, −0.54)	<0.001	878	−0.14 (−0.18, −0.09)	<0.001	
Body weight, kg	1156	−1.34 (−1.48, −1.20)	<0.001	1015	−1.61 (−1.79, −1.42)	<0.001	884	−0.35 (−0.48, −0.23)	<0.001	
FPG, mg/dL	1108	−26.6 (−29.6, −23.7)	<0.001	963	−28.3 (−31.6, −25.1)	<0.001	919	−2.0 (−4.7, 0.6)	0.014	
SBP, mmHg	1143	−3.9 (−4.8, −3.0)	<0.001	997	−3.6 (−4.6, −2.6)	<0.001	872	0.2 (−0.8, 1.3)	0.853	
DBP, mmHg	1143	−1.7 (−2.3, −1.1)	<0.001	997	−1.4 (−2.0, −0.7)	<0.001	872	0.4 (−0.3, 1.1)	0.161	
Total cholesterol, mg/dL	485	3.6 (0.6, 6.6)	<0.001	446	2.9 (−0.2, 6.0)	0.006	433	−3.2 (−6.4, 0.1)	0.586	
Triglycerides, mg/dL	587	−6.9 (−22.8, 9.0)	<0.001	542	−16.6 (−27.0, −6.2)	<0.001	501	−12.5 (−31.7, 6.7)	0.086	
LDL-C, mg/dL	797	0.58 (−1.4, 2.5)	0.009	730	1.1 (−0.9, 3.1)	0.014	635	−2.5 (−4.6, −0.5)	0.180	
HDL-C, mg/dL	400	1.1 (0.2, 2.1)	<0.001	369	2.4 (1.2, 3.7)	<0.001	371	1.6 (0.8, 2.3)	<0.001	
Notes.

Data are presented as mean (95% confidence interval).

Abbreviations BMI body mass index

DBP diastolic blood pressure

FPG fasting plasma glucose

HbA1c glycated hemoglobin

HDL-C high-density lipoprotein cholesterol;

LDL-C low-density lipoprotein cholesterol

SBP systolic blood pressure

Subgroup analyses: HbA1c and body weight

We performed analyses to examine the effects of dapagliflozin treatment on HbA1c and body weight among different subgroups. For HbA1c, a statistically significant trend of greater reduction was observed in patients with higher baseline HbA1c from baseline to 3 and 6 months (Fig. 1A). In patients with baseline HbA1c between 8–9% and those >10%, a significant greater HbA1c reduction was found at Month 6 compared with Month3. Patients who received dapagliflozin as add-on therapy had a significantly greater reduction in HbA1c (−0.82%) than those who were switched from one antihyperglycemic agent to dapagliflozin at Month 6 (−0.66%, p = 0.002, Fig. 1B). Considering patients with evaluable HbA1c data at baseline and Month 6, the proportion of patients achieving the glycemic target (HbA1c <7%) were 5% and 6% for dapagliflozin add-on and switch groups at baseline, and they subsequently increased to 23% and 15% by Month 6, respectively (Fig. S1B). When stratified by baseline BMI, no significant difference in HbA1c reduction was found across subgroups (Fig. S2A). Similarly, no difference among age subgroups was found for HbA1c reduction (Fig. S2B). When stratified by dosage of dapagliflozin, patients receive 10 mg dapagliflozin had significantly greater reduction in HbA1c (−0.74%) than those who receive 5 mg at month 6 (−0.67%, p < 0.001, Fig. S2C).

Figure 1 Changes in HbA1c at 3 and 6 months (A) by baseline HbA1c and (B) by manner of dapagliflozin initiation.

Figure 2 Changes in body weight at 3 and 6 months by baseline BMI.

For body weight, treatment with dapagliflozin showed a statistically significant trend of greater weight loss with increasing baseline BMI from baseline to 3 and 6 months (Fig. 2). Among patients with evaluable data at both Month 3 and Month 6, further weight reduction was significant at Month 6 compared with Month 3 across all BMI categories, except for those with baseline BMI ≥35. When stratified by baseline HbA1c, significantly greater weight reductions were found in patients with lower baseline HbA1c throughout the study, despite similar baseline body weight among HbA1c categories (Fig. S3A). There was no difference in weight reduction across age subgroups at Month 3, but a significant difference was found at Month 6 (Fig. S3B). Comparing data between 3 and 6 months, significant reductions in weight were found in groups aged 40–65 and 65–75 years. When stratified by dosage of dapagliflozin, patients receive 10 mg dapagliflozin had significantly greater reduction in body weight (−1.65 kg) than those who receive 5 mg at month 6 (−1.43 kg, p = 0.011, Fig. S3C).

In addition, patients were stratified into four groups according to the number of antidiabetic therapies at baseline: monotherapy, dual therapy, triple therapy, and >3 drugs combination. Across these subgroups, no significant differences were observed in both HbA1c and body weight reductions (Fig. S2D; Fig. S3D).

Relationship between changes in HbA1c and body weight

Patient distributions of changes in HbA1c and body weight after initiating dapagliflozin for 6 months were presented in scatter plots (Fig. 3). Of the 1,094 patients with evaluable data, 79.9% and 77.6% of them experienced reductions in HbA1c and body weight, respectively. Moreover, 64.0% of patients showed simultaneous reductions in both outcomes (Fig. 3A). For patients in the add-on (n = 529) and switch (n = 565) groups, 69.2% and 59.4% had a reduction in both HbA1c and body weight, respectively (Figs. 3B and 3C). In terms of clinical meaningful change, 74.4% and 74.2% of patients had at least 0.5% and 1% reduction in HbA1c for 6 months, respectively. 37.1% of patients had at least 3% weight loss for 6 months. The baseline characteristics of at least 1% reduction in HbA1c, 3% weight loss and both were show in the supplement Tables 1–3, respectively.

Baseline factors associated with dapagliflozin response in HbA1c

Baseline factors associated with dapagliflozin response in HbA1c at Month 6 were shown in Table 3. Responders and non-responders were determined by using the median of change in HbA1c (−0.60%; min, max [−6.6, 3.2]) as the cutoff.

Univariate logistic regression analysis indicated that higher baseline HbA1c, FPG, add-on dapagliflozin, use of insulin were significantly associated with dapagliflozin response in HbA1c reduction, while being on dual therapy at the time of dapagliflozin initiation was significantly associated with less HbA1c reduction. In multivariate logistic regression analysis, significant associations with dapagliflozin response in HbA1c reduction were found in patients with higher baseline HbA1c (odds ratio [OR] 2.10; 95% CI [1.79–2.47], p < 0.001), and those who received dapagliflozin as add-on therapy (OR 1.60; 95% CI [1.19–2.14], p = 0.002).

Figure 3 Scatter plots for the relationship between changes in HbA1c and body weight at Month 6 among (A) total, (B) add-on, and (C) switch populations.

Table 3 Baseline factors associated with dapagliflozin response in HbA1c at Month 6 examined by logistic regression analysis.

Parameter	OR	95% CI	pvalue	
HbA1c –Univariate	
Age (years)	0.99	(0.98, 1.01)	0.277	
Gender, female	1.20	(0.94, 1.53)	0.138	
Weight (kg)	1.00	(0.99, 1.01)	0.594	
BMI (kg/m2)	1.01	(0.99, 1.04)	0.357	
HbA1c (%)	2.12	(1.87, 2.41)	<0.001	
FPG (mg/dL)	1.01	(1.00, 1.01)	<0.001	
SBP (mmHg)	1.00	(0.99, 1.01)	0.955	
Add-on dapagliflozin	1.31	(1.03, 1.67)	0.029	
Number of antihyperglycemic drug	
Monotherapy	0.76	(0.47, 1.24)	0.273	
Dual therapy	0.66	(0.45, 0.98)	0.037	
Triple therapy	0.72	(0.49, 1.06)	0.098	
Antihyperglycemic therapy	
Insulin	1.58	(1.20, 2.07)	0.001	
HbA1c –Multivariate	
HbA1c (%)	2.10	(1.79, 2.47)	<0.001	
FPG (mg/dL)	1.00	(1.00, 1.01)	0.349	
Add-on dapagliflozin	1.60	(1.19, 2.14)	0.002	
Number of antihyperglycemic drug	
Monotherapy	1.40	(0.78, 2.50)	0.255	
Dual therapy	0.94	(0.58, 1.52)	0.804	
Triple therapy	0.95	(0.59, 1.54)	0.844	
Antihyperglycemic therapy	
Insulin	0.83	(0.59, 1.18)	0.298	
Notes.

Responders and non-responders were determined by using the median of change in HbA1c (−0.60%) as the cutoff.

Abbreviations BMI body mass index

CI confidence interval

FPG fasting plasma glucose

HbA1c glycated hemoglobin

OR odds ratio

SBP systolic blood pressure

Discontinue rate and reasons for discontinuation

Among eligible patients, the total number of discontinuation at Month 3 and Month 6 were 153 (7.8%) and 247 (12.6%), respectively. The main reasons for the discontinuation were inadequate HbA1c control (n = 83; 4.23%), intolerance (n = 46; 2.35%), or poor compliance to the current regimens (n = 9; 0.46%). The most commonly reported reasons for intolerance were genital or urinary tract infections (n = 15), frequency of urination (n = 8), and vaginal itching (n = 5).

Discussion

This multicenter retrospective study presented the first nationwide real-world evidence for dapagliflozin in Taiwan. In this cohort of 1960 Taiwanese patients with T2DM, initiating dapagliflozin was associated with significant improvements in glycemic control, body weight, and BP. Overall, the changes from baseline were −0.73% for HbA1c, −28.3 mg/dL for FPG, −1.61 kg (−2.14%) for body weight, and −3.6/−1.4 mmHg for SBP/DBP at 6 months. These clinical benefits of dapagliflozin are comparable to the efficacy data from meta-analyses of RCTs (Sun et al., 2014; Zhang et al., 2014). In addition, small changes in lipids were noted, including an increase in total cholesterol, LDL-C, HDL-C, and a decrease in triglycerides.

Several observation studies assessing the effectiveness of dapagliflozin have been published recently (Brown, Gupta & Aronson, 2017; Fadini et al., 2018; Han et al., 2018; Huang et al., 2018b; Mirabelli et al., 2019; Scheerer et al., 2016; Viswanathan & Singh, 2019; Wilding et al., 2017). Despite differences in ethnicity, location, and other clinical factors, the baseline HbA1c among studies were similar to our data, ranging 8.5%–9.5%, and many patients were already on two or three OADs, some were even using insulin. The effects of dapagliflozin on glycemic control, body weight, and BP were consistent across these studies: reduction in HbA1c ranged from 0.7%–1.5% over 6 to 12 months of observation, reduction in body weight by percentage ranged 1.75%–3.83%, and reduction for SBP and DBP were of 2.3–3.8 mmHg and 1.1–2.0 mmHg, respectively. In our study, these effects were significant 3 months after initiating dapagliflozin, and aside from the BP-lowering effect that plateaued at Month 3, we found small but significant improvements in other key metabolic parameters from Month 3 to Month 6. Also, 65% of patients showed simultaneous reductions in both HbA1c and body weight, comparable to the previous reports (Fadini et al., 2018; Han et al., 2018; Scheerer et al., 2016). These data suggested that dapagliflozin is effective in real-world practice for patients with inadequately controlled T2DM.

The baseline characteristics in our cohort, such as mean age, sex ratio, and HbA1c were similar to the South-East Asia cohort in the recent global DISCOVER study (Gomes et al., 2019). Although diabetic patients in South-East Asia (and Western Pacific region) have lower BMI than those in Western countries, the current Taiwanese study had slightly higher BMI compared with the South-East Asia data in the DISCOVER study (28.3 vs. 27.3). To explore and identify whether some baseline factors may be associated with better treatment response to dapagliflozin, subgroup and logistic regression analyses were performed. We observed a significant trend of greater HbA1c reduction in patients with higher baseline HbA1c, and the association was further indicated in multivariate logistic regression analysis. This association is consistent with findings from the literature. In general, studies suggest that patients with higher baseline HbA1c experienced a greater reduction in HbA1c with dapagliflozin treatment (Brown, Gupta & Aronson, 2017; Han et al., 2018; Hong et al., 2019; Scheerer et al., 2016), and one study indicates that a lower baseline HbA1c was associated with the achievement of HbA1c goal (<7%) (Wilding et al., 2017). Other patient characteristics that may be associated with better dapagliflozin response were male (Brown, Gupta & Aronson, 2017), younger age (<45 years old) (Wilding et al., 2017), shorter disease duration (<2–4 years), (Brown, Gupta & Aronson, 2017; Han et al., 2018; Wilding et al., 2017) and non-insulin use (Brown, Gupta & Aronson, 2017; Hong et al., 2019); however, some of these associations were not identified and other data were not available in the current study. On the other hand, baseline body weight or BMI did not influence the magnitude of reduction in HbA1c, which is similar to previous studies (Han et al., 2018; Scheerer et al., 2016; Wilding et al., 2017).

Regarding the effects of dapagliflozin by manners of initiation, we found patients who initiated dapagliflozin as add-on therapy had a significantly greater reduction in HbA1c (−0.82%) than those who were switched (−0.66%) from other OADs. Two recent studies also showed treatment with SGLT2 inhibitors (i.e., dapagliflozin or empagliflozin) as add-on therapy had a greater glucose-lowering effect than as switch therapy (Han et al., 2018; Hong et al., 2019). Notably, we found DPP4 inhibitors accounted for the majority (69.5%) of the switched agents (data not shown), a finding which is likely due to the fact that Taiwan’s NHI only covers either a DPP4 inhibitor or an SGLT2 inhibitor. A single-center retrospective study in Taiwan recently reported that patients who switched from a DPP4 inhibitor to an SGLT2 inhibitor (i.e., empagliflozin) for 6 months had significant reductions in HbA1c and body weight, whereas those who remained on a DPP4 inhibitor did not experience significant changes (Huang, Huang & Hsu, 2018a). Moreover, data from long-term clinical trials demonstrated sustained efficacy and tolerability of dapagliflozin as add-on therapy (Bailey et al., 2013; Del Prato et al., 2015; Leiter et al., 2016; Matthaei et al., 2015). Taken together, switching to an SGLT2 inhibitor from other OADs such as DPP4 inhibitors may be a suitable option for antidiabetic treatment, and add-on SGLT2 inhibition could provide better clinical benefit than switch therapy in patients already taking other OADs without adequate glycemic control (Hong et al., 2019; Huang, Huang & Hsu, 2018a).

In our study, we observed a significant trend of greater weight loss with increasing baseline BMI. This trend remained when data were calculated by the percentage of weight reduction, with about 1% weight reduction in the BMI <24 category and over 2% in those with BMI ≥30 (data not shown). Besides, we also observed a significant weight loss in patients received 10 mg dapagliflozin compared with 5 mg. This result is similar with a previous study that have demonstrate a dose-dependent reduction in body weight with dapagliflozin in Chinese patients under placebo-controlled trial conditions (Cai et al., 2018). Loss of fat mass has been known to account for approximately 70% of total weight loss observed with dapagliflozin treatment (Bolinder et al., 2012). These findings have significant clinical implications in Asians, as visceral adiposity is known to be higher in Asians than Caucasians at a given BMI, contributing to insulin resistance that leads to cardiovascular and renal complication (Lim et al., 2017; Ma & Chan, 2013). Given its effectiveness on HbA1c, FPG, body weight, and BP, dapagliflozin could serve as a promising OAD for Asian population to achieve better glycemic control and reduce future risk of cardiovascular diseases.

Since several cardiovascular outcome trials of SGLT2 inhibitors have been published, their effects on cardiorenal protection have come under the spotlight (Scholtes et al., 2019). Both large-scale RCT and real-world studies (e.g., CVD-REAL study) indicate that dapagliflozin lowers hospitalization for heart failure and cardiovascular mortality in T2DM patients with existing or risk of cardiovascular disease (Cavender et al., 2018; Norhammar et al., 2019; Raparelli et al., 2020; Wiviott et al., 2019). Although cardiovascular events were not assessed in this study, the effects of dapagliflozin on several cardiovascular risk factors (e.g., HbA1c, body weight, BP) observed were similar to those demonstrated in the DECLARE-TIMI 58 trial. In addition, change in lipid profile by dapagliflozin was similar to that reported in a pooled analysis of clinical trials (Jabbour et al., 2018). While the slightly elevated LDL-C level may be of concern, a study showed that dapagliflozin increases concentration of the less atherogenic large buoyant LDL-C, while the potent atherogenic small dense LDL-C remained suppressed (Hayashi et al., 2017). Future study with long-term follow-up would be expected to assess the cardiovascular outcomes with dapagliflozin treatment in the Taiwanese population.

Our study provided clinical effects of dapagliflozin on glycemic, weight, BP control at 6 months using a large representative cohort of patients withT2DM in Taiwan. Besides, comprehensive analyses were performed to identify subgroups and baseline predictors for better dapagliflozin response. However, there are several limitations in this study. First, due to a lack of comparison arm in prospective fashion, the reported effect size might not reflect the true differences. In addition, limited by the retrospective observational study design, partial loss of follow-up, missing data, and other confounding factors were inevitable. Due to the relatively low percentage of patients with prior CAD (7.8%), cerebrovascular disease (1.7%), and heart failure (2.6%), we were unable to further analyzed the outcomes stratified by primary and secondary prevention groups. Besides, the safety data was not prospectively collected. However, we have recorded the reasons for discontinuation due to intolerance. In addition, disease duration and renal function (eGFR) were not recorded, so we were unable to confirm their association with dapagliflozin response in HbA1c, as observed in other studies (Brown, Gupta & Aronson, 2017; Han et al., 2018; Wilding et al., 2017).

Conclusions

The LEAD study presented the effectiveness of dapagliflozin observed from the highest number of Chinese patients with T2DM in a real-world setting to date. Add-on therapy showed better glycemic control than switch therapy. The initiation of dapagliflozin was associated with significant improvements in glycemic control, body weight, and BP in Taiwanese patients, which is comparable to that in RCTs and other observational studies.

Supplemental Information

Supplemental Information 1 Proportion of patients within each HbA1c group at baseline and Month 6 for (A) total population (n=1253) and (B) by manner of dapagliflozin initiation

Click here for additional data file.

Supplemental Information 2 Changes in HbA1c at 3 and 6 months (A) by baseline BMI and (B) by age

Click here for additional data file.

Supplemental Information 3 Changes in body weight at 3 and 6 months (A) by baseline HbA1c and (B) by age

Click here for additional data file.

Supplemental Information 4 Raw data

Click here for additional data file.

Writing and editorial assistance in the preparation of this manuscript was provided by Chian-Yi Liu at EMD Asia Scientific Communication (Taiwan branch) Co., Ltd.

Additional Information and Declarations

Competing Interests

Author Contributions

Human Ethics

Data Availability

The authors declare there are no competing interests.

Jung-Fu Chen, Yung-Chuan Lu, Yi-Jen Hung, Szu-Ta Chen, Chieh-Hsiang Lu, Chwen-Yi Yang, Ching-Chu Chen and Pi-Jung Hsiao conceived and designed the experiments, performed the experiments, authored or reviewed drafts of the paper, and approved the final draft.

Yun-Shing Peng, Chung-Sen Chen, Chin-Hsiao Tseng, Pei-Chi Chen, Ting-I Lee, Yi-Sun Yang, Ching-Ling Lin, Chun-Chuan Lee and Ju-Ying Jiang conceived and designed the experiments, performed the experiments, prepared figures and/or tables, and approved the final draft.

Shih-Te Tu conceived and designed the experiments, performed the experiments, prepared figures and/or tables, authored or reviewed drafts of the paper, and approved the final draft.

The following information was supplied relating to ethical approvals (i.e., approving body and any reference numbers):

The Institutional Review Board Changhua Christian Hospital, Research Ethics Committee China Medical University & Hospital, The Institutional Review Board Chung Shan Medical University Hospital, Institutional Review Board Chia-Yi Hospital, Taipei City Hospital Research Ethics Committee, Chang Gung Medical Foundation Institutional Review Board, Kaohsiung Medical University Chung-Ho Memorial Hospital, Institutional Review Board Chi Mei Medical Center, Institutional Review Board of the E-DA Hospital, The Institutional Review Board of Shin Kong Wu Ho-Su Memorial Hospital, Institutional Review Board, Tri-Service General Hospital, Research Ethics Review Committee Far Eastern Memorial Hospital, Institutional Review Board of the Cathay General Hospital, TMU-Joint Institutional Review Board, National Taiwan University Hospital Research Ethics Committee, Institutional Review Board MacKay Memorial Hospital granted Ethical approval to carry out the study within its facilities.

The following information was supplied regarding data availability:

The raw data are available as Supplemental Files.

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
