# Peer review of "Use and effectiveness of dapagliflozin in patients with type 2 diabetes mellitus: a multicenter retrospective study in Taiwan"

_PeerJ, doi:10.7717/peerj.9998_

## Round 0.1 · original submission · Major Revisions

The reviewers have found some issues that need to be addressed and implemented in the methods, in the study design, and in the discussion.

Reviewer 1 ·

Basic reporting

In this manuscript, Jung-Fu Chen and colleagues evaluated the use and short-term effectiveness of dapagliflozin, the first‐in‐class SGLT-2 inhibitor in Taiwan since 2016, under routine clinical practice conditions. Major strengths of the study include the large sample size (1960 diabetic patients) with nationwide distribution and the assessment of baseline predictors of clinical response.

1. I would suggest the Authors to update the reference list of real-world evidences on SGLT-2 inhibitors with:
Mirabelli M, Chiefari E, Caroleo P, Vero R, Brunetti FS, Corigliano DM, Arcidiacono B, Foti DP, Puccio L, Brunetti A. Long-Term Effectiveness and Safety of SGLT-2 Inhibitors in an Italian Cohort of Patients with Type 2 Diabetes Mellitus. J Diabetes Res. 2019 Nov 4; 2019:3971060. doi: 10.1155/2019/3971060.
Also, epidemiological trends of diabetes mellitus in Taiwan should be integrated and updated with the most recent International Diabetes Federation (IDF) and 2019 Diabetes Atlas references.

2. Scatter Plots in Figure 3 are not easy and intuitive to read. I suggest the use of conditional colors and larger fonts, other than a presentation of panels ordered from A to C (and not the current B, A, C).

3. The “LEAD” acronym of this multicentric retrospective study should be explained in the introduction section, with the meaning of terms it refers to. The LEAD abbreviation could be mistaken for a well-known clinical trial program assessing the efficacy of GLP1-RA Liraglutide.

4. Table 1: the mean values of SPB and DBP should be illustrated in two separate rows.

5. Given its importance, Table S1 should be inserted as an ordinary table and not in supplementary materials.

6. I would recommend English editing to improve readability of the manuscript and solve minor grammar issues.

Experimental design

This real-world study adheres to high ethical standards and research integrity principles. However, I would rise some concerns about clarity in the Methods section.
1. It should be described how patient data were retrospectively collected and unified in a single database. Was it an automated procedure or were data extracted manually from the original medical records? The first option would have secured a higher degree of uniformity in the collection and analysis of data from multiple diabetes centers, thereby representing a strength point of this study.

2. The statistical software used for the analysis should be specified.

3. The exclusion criteria for participant eligibility are not clear. In particular, the phrase “patients undergoing treatment with other investigational drugs concurrently during the retrospective data collection period” appears confusing. Were you excluding patients concurrently treated with other novel antidiabetic agents (e.g. GLP1 RA) or patients generally involved in clinical trials independently of diabetes?

4. The methods section should include how missing data were handled for the statistical analysis.

The influence of dapagliflozin dosage (5 mg vs 10 mg) on glycemic and weight control has not been assessed. It would be an interesting point, given that the higher dose of dapagliflozin was not routinely administered in this study.

Validity of the findings

This real-world study on dapagliflozin effectiveness in Taiwanese patients covers a central topic in Diabetology, and so Health Sciences. However, given the interesting results, the Discussion section could be much richer.

1. I would suggest the Authors to discuss the paradoxical negative influence on dapagliflozin response by a dual combination therapy (Table S1, OR 0.6). It cannot be excluded that the adjunct of dapagliflozin on distinct dual combination therapy options (e.g. met+sulfonylurea, met+acarbose, met+ insulin etc) may affect its effectiveness. For example, Mirabelli et al. evidenced that the SGLT2 inhibitor-sulfonylurea combination therapy has a negative impact on weight control in owerweight/obese diabetic patients.

2. Although the primary outcome of this study is the effectiveness of dapagliflozin, the finding of a high rate of treatment interruption within 6 months of follow-up (400 patients, 20.4%) should be better detailed in the Results section and properly discussed. The “intolerance” term suggests genitourinary side effects in most cases. However, given the high prevalence of nephropathy in this study population (34.6%, Table 1) the risk of dapagliflozin-induced kidney injury is also conceivable, together with a loss in glycemic efficacy for a glycosuric agent. Also, how many patients shifted from dapagliflozin to different antidiabetic regimens due to “inadequate HbA1c control”?

Additional comments

The study is interesting and well executed, however, greater clarity in the methods description is required, together with the addition of dapagliflozin dosage in the regression logistic model to predict a positive glycemic response. Also, the discussion section should explore the study findings outlined above.

Reviewer 2 ·

Basic reporting

Chen et al. conducted a multicentre retrospective study to examine the effectiveness of dapagliflozin on cardiovascular risk factors in Taiwan. It studied 1960 people with type 2 diabetes (52% men; mean age 58 years, HbA1c 8.8%, BMI 28 kg/m2) who were initiated with or switched to dapagliflozin. At baseline, 8% and 35% had prior coronary artery disease and nephropathy, respectively. Raw data were shared.

A major pitfall in the present study was lack of a control group. The effects of dapagliflozin were in line with existing literature, which was also stated in the Discussion.

Other major comments are as follows:
1. Line 89: Please clarify what did LEAD mean.

4. Discussion:
a) Need more details of the study limitations.
b) A few references that can be useful throughout the manuscript:
https://pubmed.ncbi.nlm.nih.gov/29852973/
https://pubmed.ncbi.nlm.nih.gov/31416989/
https://pubmed.ncbi.nlm.nih.gov/31902326/
https://pubmed.ncbi.nlm.nih.gov/27866701/

Experimental design

2. Statistical analysis:
a) How did the authors handle missing data including a loss to follow-up and drug non-adherence? The results shown in Table 2 were per-protocol analysis. Would this introduce bias?

Validity of the findings

3. Results:
a) Given 90% of people were treated with metformin at baseline, did the authors compare the effects of using dapagliflozin as 2nd and 3rd line therapy, as well as in different dual, triple or >3 drug combinations?
b) Lines 158-160: What was the proportion of drug initiation/switch in primary and secondary prevention groups?
c) Lines 181-184: Were the baseline characteristics similar between people who received add-on therapy versus switching therapy?
d) Safety and non-adherence data are important and should be reported.
e) Table 1: What were the proportions of people with or without cardiovascular disease/nephropathy treated with dapagliflozin at baseline? Did they show similar effectiveness?
f) Table S1: Did the authors report the logistic regression results of other cardiovascular risk factors e.g. body weight?
g) Figure S3: The majority of results showed a greater response in people with worse glycemic control. However, the change in body weight in people with HbA1c >10% was less than those with better glycemic control. What could be the possible reasons?

---

## Round 0.2 · Major Revisions

Some issues previously raised by the reviewers still need to be addressed.

Reviewer 1 ·

Basic reporting

The manuscript has been improved, addressing this reviewer’s comments with careful attention. There is only a minor point that should still require Authors’ consideration.

Please provide these novel subgroup analyses, evaluating the real-world effects of dapagliflozin dosage (5mg vs 10 mg) on HbA1c and body weight, to the reader as a supplementary figure, with differences (p values) among subgroups at 3 and 6 months.
Subgroup stratification by dapagliflozin dosage should be cited in material and methods, and findings should be shown and briefly discussed in the appropriate manuscript sections, taking into account that a dose-dependent reduction in body weight with dapagliflozin has been demonstrated in Chinese patients under placebo-controlled trial conditions.

(Cai X. et al. The Association Between the Dosage of SGLT2 Inhibitor and Weight Reduction in Type 2 Diabetes Patients: A Meta‐Analysis. Obesity (Silver Spring). 2018 Jan;26(1):70-80. doi: 10.1002/oby.22066 ).

Experimental design

no comment

Validity of the findings

no comment

Reviewer 2 ·

Basic reporting

Thank you for the revision.

A few comments:
Line 87-92: Current professional guidelines recommend the use of glucose-lowering drugs based on compelling indications while HbA1c is an important tool in glycemic monitoring. Suggest rephrase.

Line 244-245 and throughout the manuscript, tables and figures: Odds ratios and 95% confidence intervals in 2 decimal points and P-values in 3 decimal points would have provided sufficient precision.

Experimental design

Line 157-171: The authors should state that only patients with complete data at baseline, month 3 and month 6 were included. Information on the handling of missing data is required either being described in the Statistical Analysis section or in a supplementary table.

Validity of the findings

Line 71-72 and Line 230: Please clarify the definitions of “improvement in HbA1c” and “simultaneous reduction in HbA1c and body weight”. Readers need to know if these improvements were clinically meaningful e.g. the proportion of patients with at least 0.5% or 1% HbA1c reduction; at least 3% weight loss. Based on these, we can know the characteristics and medication profiles of patients who had a very good response to dapagliflozin. Fig 3 only provided crude visualization.

Response to Comment 3 of Reviewer 2: The 2 bar charts showing the effectiveness of dapagliflozin on HbA1c and body weight by different combination therapy did report some useful data. This reviewer opined that the results should be reported.

Study limitations:
- As mentioned in the previous review, the authors should discuss the limitation related to a lack of comparison arm and thus, the reported effect sizes might not reflect the true differences.
- The inability to analyse by primary and secondary prevention groups.

---

## Round 0.3 · accepted · Accept

The authors have satisfactorily addressed all the issues raised by the reviewers.

Reviewer 1 ·

Basic reporting

no comment

Experimental design

no comment

Validity of the findings

no comment

Reviewer 2 ·

Basic reporting

No comment

Experimental design

No comment

Validity of the findings

No comment

Additional comments

Thank you for the revision. I have no further comments.